PREPARED FOR SUBMISSION TO JHEP                    MITP-20-086, ZU-TH-57/20

# Gravitational Waves from an Axion-Dark Photon System: A Lattice Study

**Wolfram Ratzinger,**[a] **Pedro Schwaller,**[a] **and Ben A. Stefanek**[b]

[a]*PRISMA+ Cluster of Excellence & Mainz Institute for Theoretical Physics,*
*Johannes Gutenberg-Universität Mainz, 55099 Mainz, Germany*

[b]*Physik-Institut, Universität Zürich, CH-8057 Zürich, Switzerland*

*E-mail:* w.ratzinger@uni-mainz.de, pedro.schwaller@uni-mainz.de,
bestef@physik.uzh.ch

ABSTRACT: In this work, we present a lattice study of an axion - dark photon system in the early Universe and show that the stochastic gravitational wave (GW) background produced by this system may be probed by future GW experiments across a vast range of frequencies. The numerical simulation on the lattice allows us to take into account non-linear backreaction effects and enables us to accurately predict the final relic abundance of the axion or axion-like particle (ALP) as well as its inhomogeneities, and gives a more precise prediction of the GW spectrum. Importantly, we find that the GW spectrum has more power at high momenta due to $2 \rightarrow 1$ processes. Furthermore, we find the degree of polarization of the peak of the GW spectrum depends on the ALP-dark photon coupling and that the polarization can be washed out or even flipped for large values thereof. In line with recent results in the literature, we find the ALP relic abundance may be suppressed by two orders of magnitude and discuss possible extensions of the model that expand the viable parameter space. Finally, we discuss the possibility to probe ultralight ALP dark matter via spectral distortions of the CMB.

## 1 Introduction

The axion as a solution to the strong CP problem of quantum chromodynamics (QCD) is a highly motivated ultraviolet (UV) extension of the Standard Model (SM) [1]. It arises as the Nambu-Goldstone boson of a spontaneously broken $U(1)$ Peccei-Quinn (PQ) global chiral symmetry that is anomalous under QCD. Effects due to QCD instantons explicitly break the PQ symmetry, generating a periodic potential for the axion with a CP-conserving minimum, thus providing a dynamical solution to the strong CP problem. In the process, the continuous shift symmetry of the axion is broken down to a discrete one and the axion acquires a mass, potentially allowing it to be identified as a cold dark matter (DM) candidate [2–4] [1]. While this particular type of axion is intimately tied to QCD, axion-like particles (ALPs) are generic features of new high energy physics. Indeed, ALPs arise

---

[1]The abundance of the QCD axion with a PQ-breaking scale $f \gtrsim 10^{12}$ GeV overcloses the universe unless the initial axion field value is unexpectedly small.

routinely from spontaneously broken global symmetries and have been proposed as e.g. a dynamical solution to the hierarchy problem [5], natural inflaton candidates [6], and fantastic cold DM candidates.

Axion-like particles also appear as a generic prediction of string theory, arising as the Kaluza-Klein zero-modes of higher dimensional anti-symmetric tensors required for anomaly cancellation [7–10]. Such ALPs are expected to have their corresponding PQ symmetries spontaneously broken at scales $f$ ranging from the grand-unification scale $M_{\mathrm{GUT}} \sim 10^{16}$ GeV to the reduced Planck scale $M_P = 2.44 \times 10^{18}$ GeV. Additionally, instanton effects which explicitly break the PQ symmetries can be exponentially small, leading to a colossal range of possible ALP masses [9]. Axion-like particles on the ultralight end of this spectrum become "fuzzy" with important implications for structure formation if they constitute all of DM [11]. [2] Interestingly, a generic string compactification is expected to result in a large number ($\gtrsim 10$) of ALPs, potentially allowing for a linear combination that couples to some gauge field with enhanced strength [12, 13].

While such ALPs are still too weakly coupled to be probed via couplings to SM fields, it was recently shown in Ref. [14] that such models can produce a large, stochastic gravitational wave (GW) signal in the early universe if the enhanced coupling is to a hidden $U(1)$ gauge boson.[3] In this case, a tachyonic instability is induced when the ALP begins to oscillate for a specific range of "dark photon" momenta controlled by the ALP mass $m$. Dark photon modes in this range have their underlying vacuum fluctutations ($\rho_{\mathrm{vac}} \sim m^4$) exponentially amplified until their energy density becomes of order that of the ALP ($\rho_{\mathrm{ALP}} \sim m^2 f^2$), a growth factor of $\mathcal{O}(f^2/m^2)$ that results in a classical, highly anisotropic dark photon energy distribution that sources GWs. Furthermore, the amplification of vacuum fluctuations occurs in a parity-asymmetric way due the non-vanishing expectation value of the parity-violating ALP-dark photon operator. As a result, the produced GW spectrum is typically highly chiral in the peak region and is expected to be a smoking gun for and indeed may be the unique probe of such models.

As dark photon production occurs at the expense of energy in the ALP field, some parameter space where the ALP relic abundance would normally overclose the universe can be opened up. However, care here is required as these dynamics also backreact on the ALP field due to inverse decay and scattering processes involving ALPs and dark photons. These processes introduce a limit to how much energy can be transferred from the ALP to dark photons, and introduce anisotropies in the initially homogeneous ALP field. Thus, linear analyses of the system such as that of Ref. [14] break down and one must perform a detailed lattice study to correctly capture the dynamics. This fact was previously pointed out in Refs. [26–28], where it was found that the ALP relic abundance can be suppressed at most by a factor of $\mathcal{O}(10^{-2})$.

In this work, we perform our own lattice study in order to further understand the non-perturbative dynamics of the system and its impact on the GW spectrum. We solve the equations of motion for the full axion, dark photon, and GW system in position space on a

---

[2]Here ultralight means $m \lesssim 10^{-18}$ eV as set by the Jeans scale of the baryons [10].

[3]Dark photons of this type can appear in UV complete axion models [15]. GW probes of axion models have also been explored in the context of phase transitions [16–20] and inflation/preheating [21–25].

discretized spacetime lattice. In particular, our implementation is based on the staggered grid algorithm of Refs. [29, 30] which ensures that the discretized theory respects all the same symmetries of the continuous one, importantly including gauge invariance and the shift symmetry of the ALP. Additionally, our entire lattice implementation reproduces the continuum version of the theory up to an error which is quadratic in the lattice spacing.

We are able to confirm previous work suggesting that the ALP relic abundance can be suppressed by roughly 2 orders of magnitude, in addition to robustly establishing the existence of the GW spectrum predicted in Ref. [14]. We find that the main changes to the GW spectrum when compared to the results of the linear analysis are: i) an enhancement of power at higher momenta due to $2 \to 1$ processes not present in the linear analysis and ii) a dependence of the polarization of the GW spectrum on the ALP-dark photon coupling $\alpha$. The second point is expected since the two dark photon helicities are coupled through the ALP, so depending on the value of $\alpha$ the polarization tends to be washed out or "frozen-in" at some value depending on when backscattering processes decouple. We discuss extensions to the original model which allow for additional suppression of the ALP relic abundance and update the viable parameter space in the $f$ vs. $m$ plane. We also comment on the possibility to probe ultralight ALPs via spectral distortions of the CMB induced by gravitational waves.

## 2 Model Review

Here, we give a brief overview of the Audible Axion model introduced in Ref. [14]. The original simplified model consisted of an axion field $\phi$ and a massless dark photon $X_\mu$ of an unbroken $U(1)_X$ Abelian gauge group

$$\mathcal{S} = \int d^4x \sqrt{-g} \left[ \frac{1}{2} \partial_\mu \phi \, \partial^\mu \phi - V(\phi) - \frac{1}{4} X_{\mu\nu} X^{\mu\nu} - \frac{\alpha}{4f} \phi X_{\mu\nu} \widetilde{X}^{\mu\nu} \right], \qquad (2.1)$$

where the parameter $f$ is the scale at which the global PQ symmetry corresponding to the Nambu-Goldstone field $\phi$ is spontaneously broken. The dark photon field strength is $X_{\mu\nu}$ with $\widetilde{X}^{\mu\nu} = \epsilon^{\mu\nu\alpha\beta} X_{\alpha\beta}/2$ its dual [4]. The strength of the axion-dark photon coupling is parameterized by $\alpha$, which in general can be larger than the fundamental $U(1)_X$ coupling [5]. We also assume the PQ symmetry is explicitly broken at the scale $\Lambda \sim \sqrt{mf}$, generating the potential $V(\phi)$, a mass $m$ for the axion, and breaking the continuous shift symmetry of the ALP down to a discrete one, $\phi \to \phi + 2\pi n$. The potential should be invariant under this discrete shift symmetry, thus for simplicity we choose

$$V(\phi) = m^2 f^2 \left( 1 - \cos \frac{\phi}{f} \right), \qquad (2.2)$$

unless otherwise specified.

---

[4] Our convention is $\epsilon^{0123} = 1/\sqrt{-g}$

[5] We consider $\alpha > 1$ in order to have efficient particle production, which can be obtained in several UV completions, see e.g. [12, 13, 31].

We limit our analysis to the case of a massless dark photon, which allows us to work in temporal gauge $X_0 = 0$. In an expanding background $ds^2 = a^2(\tau)(d\tau^2 - d\boldsymbol{x}^2)$, the equations of motion governing the system are

$$\phi'' + 2aH\phi' - \boldsymbol{\nabla}^2\phi + a^2 V'(\phi) - \frac{\alpha}{fa^2}\boldsymbol{X}' \cdot \left(\boldsymbol{\nabla} \times \boldsymbol{X}\right) = 0\,, \tag{2.3}$$

$$\boldsymbol{X}'' + \boldsymbol{\nabla} \times \left(\boldsymbol{\nabla} \times \boldsymbol{X}\right) + \frac{\alpha}{f}\left[\phi'(\boldsymbol{\nabla} \times \boldsymbol{X}) - \boldsymbol{\nabla}\phi \cdot \boldsymbol{X}'\right] = 0\,, \tag{2.4}$$

where primes denote derivatives with respect to conformal time $\tau$ and $H = a'/a^2$ is the Hubble rate. Additionally, one has the Gauss constraint

$$\boldsymbol{\nabla} \cdot \left[\boldsymbol{X}' + \frac{\alpha}{f}\phi\left(\boldsymbol{\nabla} \times \boldsymbol{X}\right)\right] = 0\,. \tag{2.5}$$

We assume the PQ symmetry is broken before the end of inflation $f > H_I$, leading to an axion field that is spatially homogeneous over the visible universe. The initial field value of the axion is drawn from a uniform random distribution $\theta = \phi_0/f \in [-\pi, \pi]$, where $\theta \sim \mathcal{O}(1)$ is the initial misalignment angle. While $H > m$ is satisfied, Hubble friction is important and the axion field is overdamped, thus the initial velocity tracks the slow-roll attractor. As is well known, massless vector modes are not excited during inflation so we take the dark photon to be in the Bunch-Davies vacuum initially. We further assume that the universe is radiation-dominated with the axion contributing sub-dominantly to the total energy density.

With these initial conditions, one can study the axion-dark photon system by initially neglecting any spatial dependence of the axion $\phi(\tau, \boldsymbol{x}) \to \phi(\tau)$. In this limit, the equation of motion for the dark photon in momentum space becomes

$$X''_\pm(\tau, \boldsymbol{k}) + \left(k^2 \pm k\frac{\alpha}{f}\phi'(\tau)\right) X_\pm(\tau, \boldsymbol{k}) = 0\,, \tag{2.6}$$

where $X_\pm$ are the mode functions of the two circular polarizations of the dark photon. This modification of the dispersion relation leads to the modes $k \sim \alpha|\phi'|/(2f)$ of the polarization $-\text{sgn}(\phi')$ experiencing a tachyonic instability once $H$ drops below $m$ and the axion starts to freely oscillate. Due to this instability, the energy in the dark photon quickly grows from the vacuum value $k^4 \sim m^4$ to an $\mathcal{O}(1)$ fraction of the axion energy $\propto m^2 f^2$. At this point, one expects a backreaction of the dark photon onto the axion dynamics and for the axion field to develop anisotropies. Thus, one must study the system on the lattice in order to correctly capture the dynamics. Throughout this work, it will be useful to compare the case where the axion is treated as a homogeneous field as in Eq. (2.6) and Refs. [14, 32, 33] to the fully general lattice study. We thus define the **linear analysis** as the case where the axion is treated as a homogeneous field, valid before the dark photon backreacts on the axion dynamics.

## 3   Lattice Formulation and Validation

We solve the full equations of motion of the coupled axion and dark photon system by discretizing space and time. To ensure that we recover the correct theory in the continuum

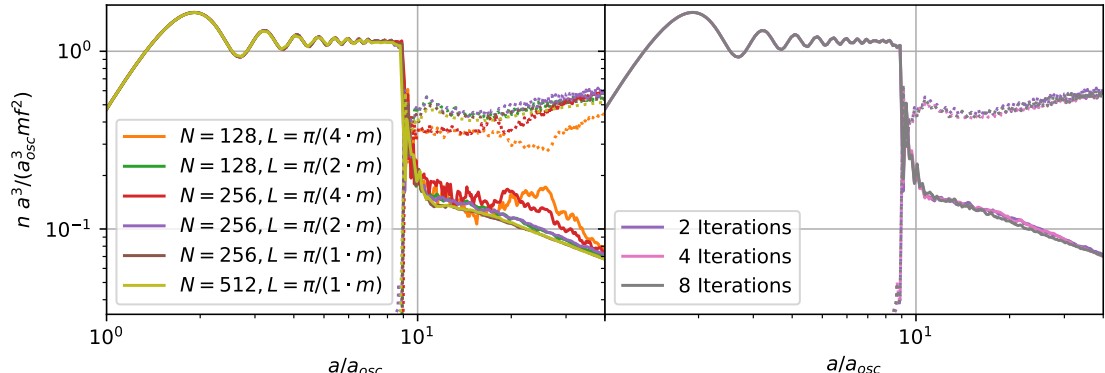

**Figure 1**. Comoving axion (solid) and dark photon (dotted) number densities for different choices of the lattice parameters with $\alpha = 60$ and $\theta = 1$ held fixed. In the left panel, $L$ and $N$ are varied while the number of iterations in the implicit scheme is held fixed at 2. Similarly, the right panel fixes $N = 256$, $L = \pi/(2m)$ and varies the number of iterations in the implicit scheme. The different choices agree to within $\sim 10\%$ except in the case of the smallest length $L = \pi/(4m)$.

limit, the discretized theory must have the same symmetry structure as the continuum one. Ideally, the discretization should reproduce the continuum theory up to an error which is high order in the lattice spacing to ensure fast convergence. Our implementation meets the following requirements:

- The continuum versions of the equations are reproduced up to $\mathcal{O}(dx_\mu^2)$, where $dx_\mu$ denotes the spatial and temporal distance between lattice sites.

- The discretization admits gauge invariance.

- The shift symmetry $\phi \to \phi + \epsilon$ of the continuum theory is respected on the lattice. This is equivalent to the discretized version of $X_{\mu\nu}\widetilde{X}^{\mu\nu} = \partial_\mu(2X_\nu\partial_\alpha X_\beta\epsilon^{\mu\nu\alpha\beta})$ being a total (lattice) derivative.

We implement these features using a staggered grid algorithm closely following Refs. [29, 30]. The equations of motion for the transverse-traceless metric fluctuations are solved to obtain the GW spectrum following Refs. [34, 35], where an algorithm is implemented that also reproduces the continuum up to $\mathcal{O}(dx_\mu^2)$.

We simulate a comoving volume $L^3$ with side length $L = \pi/m$ and $N = 512$ lattice sites along each direction with periodic boundary conditions such that we cover comoving momenta $2 \leq k/(ma_{\rm osc}) \leq 512$. This comfortably covers the range of momenta experiencing tachyonic growth, $k \sim \theta\alpha ma_{osc}/2$ for $\theta = \mathcal{O}(1)$ and $\alpha \sim 40 - 100$ [6]. The lattice parameters are thus $L$, $N$, and the number of iterations in the implicit scheme. We varied these parameters to ensure that none of our results depend on them, see Fig. 1 where we

---

[6]For benchmark points with $\theta = 3$, a smaller box $L = \pi/(3m)$ was used in order to resolve the UV dynamics properly. When attempting to capture the late time behavior of the axion abundance in Fig. 4, we used $N = 128$ and $L = \pi/(2 \cdot m)$ as the simulation must be run longer.

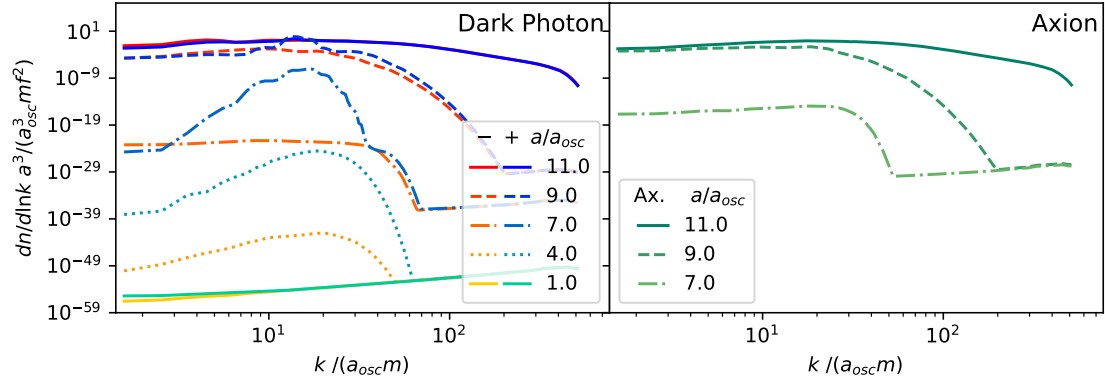

**Figure 2**. Early evolution of the dark photon (left) and axion (right) spectra. The model parameters are $\alpha = 60$ and $\theta = 1$. The "+" polarization is the first to experience tachyonic instability.

show the evolution of the axion and dark photon number densities for different lattice parameters. To keep the computational cost down, we only go to second order (2 iterations) in the implicit scheme used to solve the equations of motion as justified in the right panel of Fig. 1. For a detailed description of the lattice numerics, see Appendix B.

## 4 Lattice Results

The lattice simulation was performed with $m = 10^{-2}$ eV and $f = 10^{17}$ GeV held fixed for all runs. We then use the scaling relations described in Section 4.2 to adapt the results to other values of the model parameters. In the left panel of Fig. 2, we show the early evolution of the comoving dark photon number density for $\alpha = 60$ and $\theta = 1$, where the linear analysis holds. We define the start of oscillations $a = a_{\rm osc}$ by the condition $H = m$, with the dark photon initially in the Bunch-Davies vacuum such that $dn/d\ln k \propto k^3$.

At the second time step $a/a_{\rm osc} = 4$, the dark photon spectra perfectly agrees with the expectation from the linear analysis: during the first period of oscillation we have $\phi' \approx \theta m f a_{\rm osc}$ and therefore according to Eq. 2.6 the modes in the range $k \in [0, \alpha|\phi'|/f] \approx [0, \alpha\theta m a_{\rm osc}]$ experience a tachyonic instability. These are indeed the modes that are enhanced at $a/a_{\rm osc} = 4$ compared to the Bunch-Davies vacuum. In the first half period of oscillation, the axion velocity does not change sign and therefore only one helicity experiences tachyonic growth. Without loss of generality, we label the first helicity to experience tachyonic growth as "+" throughout this work. In the second half period, the "−" polarization is excited. However, the damping of the axion velocity due to Hubble friction results in a smaller range of tachyonic modes. Since the growth rate depends exponentially on the axion velocity, the amplitude of the "−" polarization is exponentially suppressed compared to the "+" polarization.

In the next time step at $a/a_{\rm osc} = 7$, we see the position of the peak move towards lower momenta. This is expected since the axion velocity is further decreased by Hubble friction. Additionally, we see a second contribution to the dark photon spectrum appearing that is

plateau shaped and falls off at an $O(1)$ multiple of the original peak momentum. Looking over to the right side of Fig. 2, we note that the appearance of this plateau happens at the same time as inhomogeneities in the axion field arise with a similar spectrum. From a particle point of view the origin of this feature is clear, as the axion-dark photon coupling allows for the (back-)scattering of two photons into an axion. The kinematics of this process dictate that the resulting spectrum should fall off at twice the dark photon peak momentum, which is what we observe. The plateau in the dark photon spectrum arises from further backscattering of dark photons into finite momentum axions and is expected to be unpolarized. The next time step at $a/a_{osc} = 9$ was chosen such that $\rho_X = \rho_\phi/2$ where we see the peak from tachyonic growth and the plateau from backscattering becoming comparable in size. The UV cutoff of the plateau also moves toward higher momenta and becomes less steep, which in the particle picture results from multiple scattering processes becoming more important as the number densities grow.

The last time step at $a/a_{osc} = 11$ is some time after the two energy densities become comparable in size. Before we take a closer look at the evolution during this period, let us make two technical comments. We chose a vanishing initial spectrum for the axion which stays zero during the first two time steps to within working precision. In general, the initial axion spectrum would depend on the inflation history. However, the axion spectrum resulting from backscattering processes is uncorrelated with and can be simply added to any initial spectrum that might exist from inflation. The second point concerns the UV behavior of the spectra at $a/a_{osc} = 7, 9$. This behavior corresponds to rounding errors due to the fact that we are dealing with field amplitudes differing by $\log_{10}(f/m) = 29$ orders of magnitude while using double precision floats with a precision of only 16 orders of magnitude. One expects the errors to take a random value in position space, uncorrelated from site to site. We have checked that this results in the UV part of the spectrum behaving as $\propto k^3$ in momentum space.

Fig. 3 shows a close up of the last two time steps from Fig. 2 as well as the final spectra taken at $a/a_{osc} = 200$. Also shown is the evolution of the spectrum of gravitational waves. The close up reveals that at $a/a_{osc} = 9$ when $\rho_X = \rho_\phi/2$, the dark photon spectrum is still dominated by the sharp, polarized peak resulting from the tachyonic instability. This initial peak and its polarization are however quickly washed out through scattering effects, resulting in a flat, unpolarized plateau. The UV cutoff of the plateau behavior is extended to slightly higher momenta after the two energy densities become comparable due to multiple scattering processes. Interestingly, another peak at lower momenta appears in the final spectrum that is dominated by the "−" polarization. We believe this peak, also present in the study of Ref. [28], is due to the tachyonic enhancement that occurs as the axion zero mode settles down to the minimum with roughly constant velocity. The axion velocity at this point is already significantly reduced by the production of dark photons and the resulting peak is therefore at smaller momenta.

## 4.1 Relic Abundance Suppression

Shortly after the dark photon energy density becomes comparable to that of the axion, the axion velocity becomes too small to allow for efficient production of dark photons

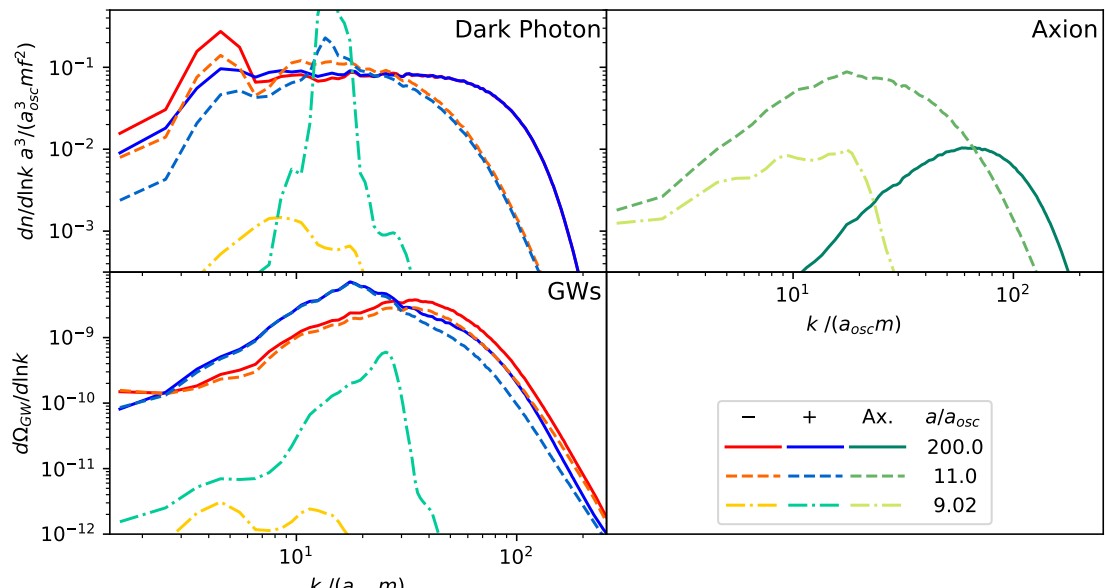

**Figure 3**. Evolution of the spectra in the non-linear regime. The model parameters are $\alpha = 60$ and $\theta = 1$. The "+" polarization is the first to experience tachyonic instability. The dark photon and axion panels correspond to those in Fig. 2 but within a much smaller range of energy densities.

through the tachyonic instability. In the linear analysis, dark photon production continued nonetheless due to a narrow parametric resonance resulting from the coherent oscillation of the homogeneous axion field. This effect could lead to a suppression of the axion relic abundance by more than 10 orders of magnitude relative to the case without any particle production.

On the lattice however, we see the axion spectrum right after the energy densities become comparable at $a/a_{osc} = 11$ has a broad peak as shown in Fig. 3. At late times, this peak moves to slightly higher momenta (similar to the dark photon), while IR power is suppressed. Low momentum axions correspond to nearly homogeneous field configurations in position space and it therefore seems plausible that the suppression of the axion abundance at low momenta is due to a parametric resonance. However, it is clear that the axion abundance at high momenta is not suppressed and that high momentum axions are still being produced at late times. This severely limits the amount by which the total axion abundance can be suppressed.

In particular, we find that the relic abundance suppression relative to the case without particle production is typically limited to $10^{-2}$, in good agreement with Ref. [26]. This can be seen clearly in Fig. 4, we show the evolution of the comoving axion energy density as calculated on the lattice compared the result from the linear analysis. They start to differ shortly after the initial backreaction, when the linear analysis predicts a much stronger depletion of the axion abundance due to the parametric resonance driven by the zero-momentum condensate. On the lattice, the axion abundance is dominated by relativistic axions, so the axion energy density scales as radiation until their momenta drops below

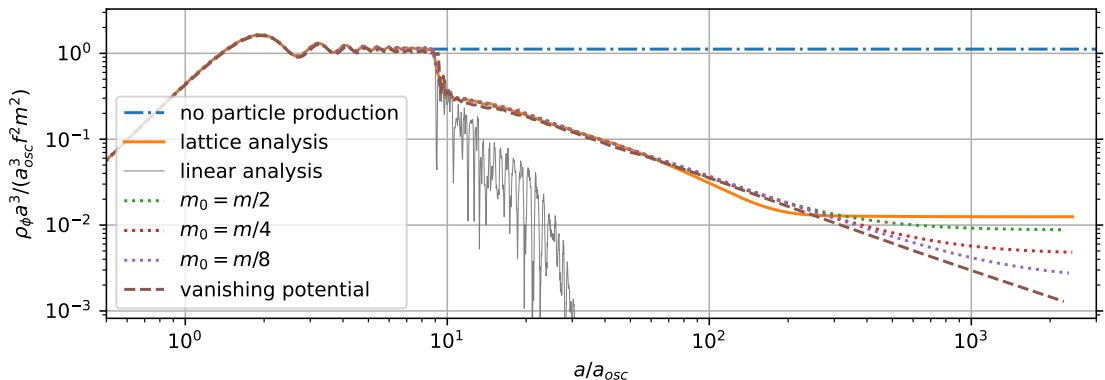

**Figure 4**. Evolution of the comoving axion energy density for $\theta = 1$. Around $a = a_{osc}$, the axion starts oscillating and scaling like matter $\rho_\phi \approx a^{-3} m^2 f^2$. Without particle production, this scaling would persist (blue dot-dashed line) yielding the standard abundance from misalignment. For $\alpha = 60$, the backreaction of dark photon production becomes strong around $a/a_{osc} \sim 9$. The thin gray line shows the result from the linear analysis, while the solid orange line gives the lattice result. The lattice result shows a suppression of the final axion abundance by $\approx 10^{-2}$ compared to the case with no particle production, in stark contrast to the linear analysis which suggests a much stronger suppresion. The dotted lines show possible further suppression in case where the final mass is adiabatically reduced, while the brown dashed line corresponds to a time dependent potential that vanishes around $a/a_{osc} = 100$ (see Sec. 5 for details).

the axion mass, locking in a suppression of about $10^{-2}$ compared to the scenario without particle production.

As shown in Fig. 5, we find that the amount of suppression has only weak dependence on $\theta$ and $\alpha$ in the regime where dark photon production is efficient ($\theta\alpha \gtrsim 30$) and friction from particle production does not cause the axion to slow-roll ($\theta\alpha \lesssim 200$). In Ref. [26], a similar study was performed in the QCD axion case (where the axion mass posses a time dependence) that comes to roughly the same conclusion. The lattice computation results in a more predictable relic abundance compared to the linear analysis, where the final abundance depended chaotically on the initial conditions [33]. Since an axion overabundance limits the parameter space with detectable gravitational waves, we discuss two potential paths to further suppress the axion abundance in Sec. 5.

## 4.2 Gravitational Wave Spectrum

Since the gravitational wave spectrum is dominantly produced in the short period after the energy densities of the axion and dark photon become comparable, the main features of the GW spectrum computed in the linear analysis of Ref. [14] survive on the lattice. In particular, the linear analysis leads to the expectation that the GW signal resulting from a polarized vector carries the same polarization as its source. Looking at the bottom panel of Fig. 3, we see that the GW spectrum is indeed strongly polarized at $a/a_{osc} = 9$, since up to this point the anisotropic stress is dominated by the highly polarized dark photon. On the lattice, we are now consistently including the axion scalar perturbations as a GW

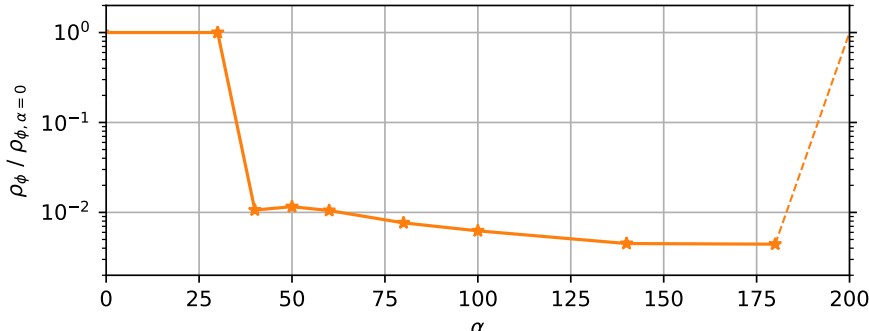

**Figure 5**. Suppression of the axion relic abundance for different values of $\alpha$ and fixed $\theta = 1$ compared to the standard misalignment case where $\alpha = 0$ and there is no dark photon production. We see that $\theta\alpha \gtrsim 30$ is required for efficient dark photon production. For values of $\theta\alpha \gtrsim 200$, friction from particle production causes the axion to slow-roll and behave as vacuum energy, thus it will quickly come to dominate the energy density of the universe. As we ignore the effect of the axion-dark photon system on the gravitational background, this regime is beyond the scope of our simulation, and we simply sketch the expected sharp loss of suppression in this region with the dashed line.

source. This can lead to a washout of polarization in the final spectrum, although as we will see some parts of the GW spectrum can remain strongly polarized.

In Ref. [14], we presented some basic scaling relations which allow for the estimation of the peak amplitude and frequency of the GW spectrum via naive dimensional analysis (NDA)

$$k_{\text{peak}} \sim 2k_* \approx \theta\alpha m \sqrt{\frac{a_{osc}}{a_*}}\, a_{osc}$$

$$\Omega_{\text{GW}}(k_{\text{peak}}) = c_{\text{eff}}\,(\Omega_\phi^*)^2 \left(\frac{a_*H_*}{k_*}\right)^2 = \frac{c_{\text{eff}}}{9}\left(\frac{f}{M_P}\right)^4\left(\frac{\theta}{\alpha}\right)^2\frac{a_*}{a_{osc}}, \tag{4.1}$$

where $c_{\text{eff}}$ is a factor quantifying the efficiency of GW emission and stars denote the corresponding quantity at the time of the initial backreaction $t_*$ where the GW spectrum is dominantly produced. Up to this time, the linear analysis roughly holds and $t_*$ can be calculated from the analytic approximations found in Ref. [36], see Appendix A for details.

In Figs. 6 and 7, we show the GW spectrum computed on the lattice for several values of $\theta$ and $\alpha$, where the NDA prediction from the scaling relation Eq. (4.1) with $c_{\text{eff}} = 1$ is indicated by a green cross. We report a final GW spectrum at $a/a_{osc} = 40$ at which point the GW signal has fully converged for all choices of the model parameters. Also shown is the spectrum at the end of the perturbative phase $t = t_*$ when $\rho_X = \rho_\phi/2$ for the first time. We see that the NDA scaling relation predicts the peak of the spectrum at $t = t_*$ to within a factor of 2, but in general fails to predict the peak of the final spectrum [7]. We suspect that $2 \to 1$ scattering processes in the phase $t > t_*$ are prolonged for large

---

[7]For large $\theta \sim 3$, the scaling relation also differs from the early spectrum because the approximation of the cosine potential as quadratic fails, invalidating the analytic solution found in Ref. [36].

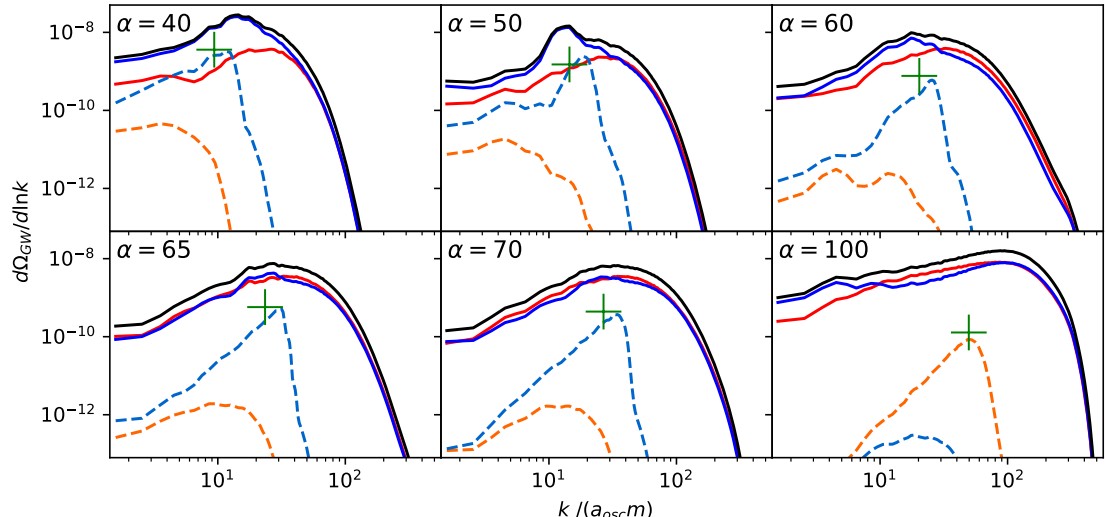

**Figure 6**. Gravitational wave spectra computed on the lattice for different values of $\alpha$ with $\theta = 1$ held fixed. The light dashed lines show the two polarizations (red, blue) when $\rho_X = \rho_\phi/2$ (roughly the end of the perturbative regime). The solid lines are the final spectra taken at $a/a_{osc} = 40$ when the GW spectrum has fully converged. The solid black line gives the sum of the two polarizations in the final spectrum and green crosses mark the NDA scaling relation from Eq. 4.1 with $c_{\text{eff}} = 1$. The source material includes the final spectra in tabulated form.

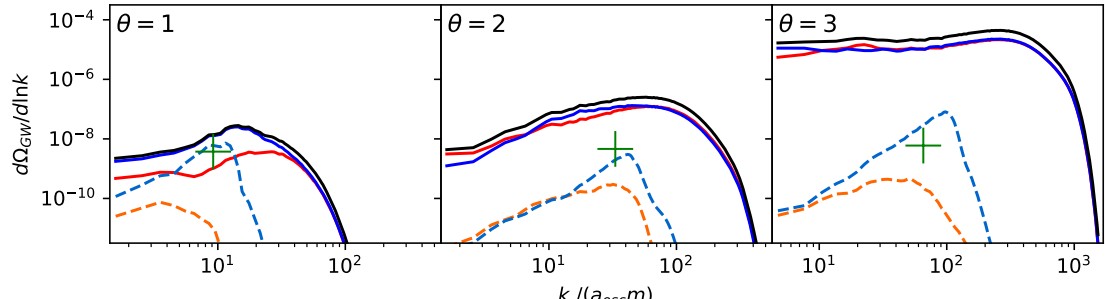

**Figure 7**. Same as Fig. 6 except $\alpha = 40$ is held fixed while $\theta$ is varied. In the case of $\theta = 3$ we chose a smaller sized box $L = \pi/(3ma_{osc})$ to better resolve the UV part of the spectrum.

values of $\theta$ and $\alpha$, leading to larger signal amplitudes and peak momenta. These processes also tend to smooth out and broaden the dark photon and axion spectra, which in turn leads to the appearance of a softened UV cutoff in the GW spectrum, as compared to the rapid exponential falloff we found in the linear analysis. The IR behavior for modes $k/(ma_{\text{osc}}) \lesssim 1$ with wavelengths larger than the lattice size $L$ is expected to approach $k^3$ scaling from causality.

Another important difference between the linear and lattice studies is that while the peak of the GW spectrum at the end of the perturbative phase $t_*$ is highly polarized, the

polarization of the peak of the final spectrum on the lattice shows a strong dependence on $\theta$ and $\alpha$. In particular, we see the polarization of the final spectrum is diminished for $\theta\alpha \gtrsim 60$. For $\theta\alpha \lesssim 60$ the GW amplitude grows by a factor of $\lesssim 10$ in the late stages $t > t_*$, while for $\theta\alpha \gtrsim 60$ the final spectrum can surpass the spectrum at $t_*$ by up to 3 orders of magnitude. The fact that the peak is largely unpolarized in cases where it is predominantly sourced after $t_*$ fits well with our earlier observation that the polarization in the dark photon spectrum is washed out after $t_*$ due to backscattering processes coupling the two dark photon helicities. The unpolarized dark photon and axion spectra thus lead to unpolarized gravitational waves. A similar suppression of polarization for large coupling constants $\alpha$ has been observed in models of natural steep inflation [37], while a study that appeared during the completion of this work found that the final polarization is limited to 10% roughly independent of $\theta$ and $\alpha$ [28]. That study considered $40 \leq \theta\alpha \leq 60$, which is the region where, in contrast, we find up to 90% polarization in the peak region. In addition, while the peak amplitude and momentum agree with our findings within roughly a factor of two, the overall shape of the spectra show significant differences.

As a final point, for $\theta\alpha \gtrsim 100$, the backreaction becomes sizeable within the first period of oscillation and the regimes of tachyonic growth and non-perturbative interaction of fluctuations are not well separated. This leads to the initially subdominant helicity surpassing the dominant one already by $t = t_*$ in the case of $\theta = 1$, $\alpha = 100$ and some strongly polarized features in the IR tail of the final spectra.

## 5 Model Extensions

As previously discussed, the axion relic density can be suppressed by only two orders of magnitude via production of dark photons once inhomogeneities in the axion field are taken into account. Overproduction of DM thus renders a sizeable part of the parameter space leading to detectable gravitational waves inconsistent with cosmology. Solutions which simply reduce the initial axion abundance such as tuning the initial misalignment angle are inappropriate in our case, as they also suppress the GW source. Instead, a mechanism is needed that reduces the axion abundance once the tachyonic phase of dark photon production (responsible for the majority of the GW signal) has ended. This could be achieved if the axion potential is in some way time-dependent or flattens out around the minimum. In both cases, the axion mass can be suppressed at late times. Let us first explore the latter scenario in the context of a monodromy-inspired potential [38–43]

$$V(\phi) = \frac{1}{2}m^2 f^2 \left(\frac{\phi}{f}\right)^2 - m_w^2 f_w^2 \left[1 - \cos\left(\frac{\phi}{f_w}\right)\right],\tag{5.1}$$

where the first term corresponds to Eq. (2.2) expanded to quadratic order in $\phi/f$, and we take $f_w < f$. Expanding for small $\phi$, the ALP mass at late times is given by

$$m_0^2 = m^2 - m_w^2,\tag{5.2}$$

which can be small if $m_w \sim m$. Defining $m_w^2 = m^2(1 - \epsilon^2)$ with $\epsilon \ll 1$ and $\varphi = \phi/f$, we can write

$$\frac{V(\varphi)}{m^2 f^2} = \frac{1}{2}\varphi^2 - \frac{f_w^2}{f^2}(1 - \epsilon^2)\left[1 - \cos\left(f\frac{\varphi}{f_w}\right)\right] . \tag{5.3}$$

In this form, we can easily see that when $\varphi \sim \theta \sim \mathcal{O}(1)$, the argument of the cosine term is large and its overall contribution to $V$ is suppressed by $f_w^2/f^2$. Thus, ALP dynamics in this regime are controlled by the $\varphi^2$ term and the axion mass is approximately $m$. However, once the ALP amplitude becomes of order $\varphi \lesssim f_w/f$, we can expand the cosine and see that the ALP mass changes from $m$ to the final mass $m_0$. Our simulations confirm that during this process the axion number density is conserved to a good approximation, leading to a suppression of the axion relic abundance which is linear in the ratio $m_0/m$ as shown in Fig. 4.

A similar setup was considered in [44, 45], which relied on the anharmonic part of the potential for self-resonant axion (and GW) production. In that case, taking $\epsilon$ small necessarily leads to a weak resonance unless the initial axion field value is very large. As we rely on the axion-dark photon coupling for particle production (which simply requires a non-vanishing $\phi'$), this incompatibility does not hold here. Indeed, the model given by Eq. (5.1) combined with a strong axion-dark photon coupling leads to sizeable GW production even for $\phi_0/f \sim 1$. We estimated in Ref. [14] that tachyonic production stops once the scale factor has grown by $a/a_{osc} = (\alpha\theta/2)^{2/3}$. Since the axion amplitude damps at least as fast as $a^{-3/2}$ (it falls off even faster when including friction from particle production), one finds

$$\frac{1}{f_w} \gtrsim \frac{\alpha}{2f} , \tag{5.4}$$

is required in order to have tachyonic particle production complete before the cosine substructure is resolved. Interestingly, this suggests a possible embedding of the model into a monodromy construction where the axion couples to the dark photon as $f_w^{-1}$, with different UV origins for the quadratic and cosine terms in Eq. (5.1), as in Refs. [46, 47]. Large $\alpha$ in such a construction could be understood in terms of the separation of scales $f/f_w$.

Another way to reduce the axion relic abundance is via a time-dependent potential. One possibility is that the axion mass at early times comes dominantly from a potential induced via $U(1)_X$ monopoles through the Witten effect [48, 49]. In this case, the axion potential is proportional to the monopole number density and thus decays as $a^{-3}$.

Finally, one could entertain the possibility that the axion is exactly massless at late times [50]. This would occur if the axion potential arises from some QCD-like dynamics, where the dark quarks temporarily acquire mass from the VEV of a dark Higgs field that later vanishes [51]. In such a case, the late time axion potential vanishes in exactly the same way as in QCD with one massless quark, and the axion relic abundance is subject only to $N_{\text{eff}}$ constraints.

## 6 Phenomenology

In Fig. 8, we show an updated overview of the model parameter space as studied in Ref. [32]. We include the regions that result in detectable GW signals as well as cosmological bounds

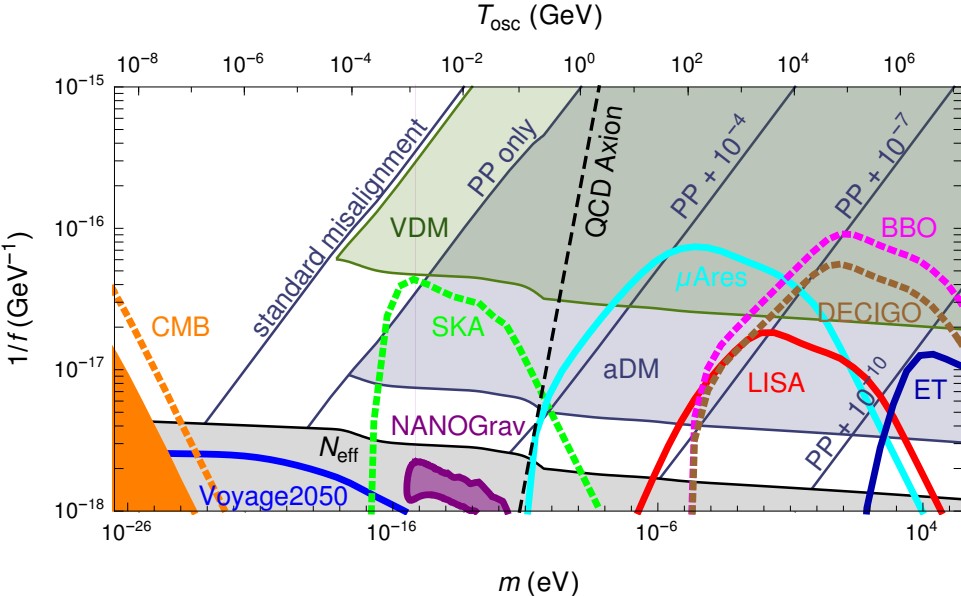

**Figure 8**. ALP parameter space in the mass vs. inverse decay constant plane with $\alpha = 100$ and $\theta = 1$ held fixed. The parameter space below the bright colored curves could be probed by future GW experiments, such as pulsar timing arrays (SKA) as well as space- (LISA, DECIGO, BBO, $\mu$Ares) and Earth-based (ET) interferometers. The filled orange region corresponds to the present limits from Planck+BICEP2+Keck and the dashed line shows the possible improvement by the LiteBird mission. The blue curve is the limit on CMB spectral distortions that could be probed by the Voyage2050 mission. The purple region is where the model could account for the recently reported NANOGrav signal. The gray region is excluded in case of a relativistic dark photon by bounds on $N_{\rm eff}$, while in the green region a massive dark photon can be a viable DM candidate. The solid diagonal lines refer to axion dark matter scenarios in which, from left to right, there is no particle production (standard misalignment), only the suppression from particle production $\approx 10^{-2}$ (PP only), or further suppression $\eta$ from model extensions (PP + $\eta$). In the blue shaded area, the axion is cool enough to be DM, assuming sufficient suppression of the relic abundance.

on the model for fixed $\alpha = 100$ and $\theta = 1$. The GW detectability curves were computed using the GW spectrum obtained from the lattice, with the IR scaling for $k \lesssim ma_{\rm osc}$ taken to be $\propto k^3$ as expected from causality. Furthermore, we use the improved scaling relations from Appendix A to calculate the axion and dark photon relic abundance.

Varying the ALP mass gives detectable GW signals across a vast range of frequencies, from the earth-based Einstein Telescope (ET) laser interferometer to the space-based interferometers LISA, BBO, DECIGO and $\mu$Ares as well as the current and future pulsar timing arrays NANOGrav and SKA. For NANOGrav, we show the $2\sigma$ region where the model could explain the recently observed signal [52]. At even lower masses, gravitational waves from ALPs can cause spectral distortions in the CMB. The solid blue curve shows the parameter space testable by the Voyage2050 mission that will be able to probe these spectral distortions at the $10^{-9}$ level [53]. For even smaller ALP masses, the bounds on CMB B-mode polarization induced by gravitational waves from Planck+BICEP2+Keck are already able to constrain the model [54]. We also show the possible improvement of

these bounds by the LiteBird mission [55].

The blue shaded region in Fig. 8 corresponds to the parameter space where the axion possibly comprises all of DM. The left diagonal bound of the region matches the dark matter abundance assuming a suppression of two orders of magnitude from particle production. The region near this line, where no further suppression is need, can be probed by SKA for $m \sim 10^{-16} - 10^{-14}$ eV and $f \sim 5 \times 10^{16}$ GeV. As discussed in Section 4, the axion transitions from the condensate into non-zero momentum states in the process of dark photon production. Axion dark matter can therefore be warm in this scenario. Requiring axion dark matter to be cool enough to form structures gives the lower bound on the blue shaded region. Observable GW signals in the space (ground)-based interferometers require an additional suppression of the axion abundance by 4 to 7 (10) orders of magnitude in order to avoid overclosure. As discussed in Section 5 this can be achieved in simple extensions of our model.

In the case where the dark photon has a sufficiently small mass such that it is relativistic at late times, it contributes to the number of effective relativistic degrees of freedom $N_{\mathrm{eff}}$. Requiring the $N_{\mathrm{eff}}$ bounds to be satisfied leads to the gray shaded exclusion region in Fig. 8. We find that the bounds from $N_{\mathrm{eff}}$ are in tension with the NANOGrav signal originating from this model, and similarly for any spectral distortions that might be probed by the future Voyage2050 mission. Although there has been recent interest in similar models with ultralight scalars and their GW signals in the context of the Hubble tension [56] as well as Quintessence [57], none of these studies incorporate the scalar perturbations in a consistent manner. Their inclusion might considerably strengthen the bounds from CMB fluctuations and therefore lead to a non-trivial probe of the model via CMB spectral distortions. If the dark photon mass is larger but still less than the axion mass in order to not interfere with the tachyonic production, the dark photon can be a viable vector dark matter (VDM) candidate [27, 58–60] in the green shaded region of Fig. 8. The origin of the lower bound is again where the dark photon DM would be too warm to be compatible with structure formation.

While non-planar networks of GW interferometers are inherently sensitive to the polarization of gravitational waves, planar interferometers and pulsar timing arrays are also sensitive to polarization in the case where there are anisotropies in the GW background, such as the one introduced by peculiar motion [61, 62]. This may offer an opportunity to distinguish the partially polarized spectrum of this model from other unpolarized signals, especially for parameter points where $\theta\alpha \lesssim 60$, where the signal is more than 90% polarized.

## 7 Discussion and Conclusions

The nature of dark matter and how it is produced in the early universe remains a mystery. Axions or ALPs are viable candidates, and coupled to a dark photon they can induce a tachyonic instability, efficiently transferring energy from the axion to the dark photon and thereby widening the viable parameter space for ALP dark matter [33]. In Refs. [14, 32]

we showed that this process can be accompanied by the production of a stochastic GW background, rendering the model testable for large decay constants.

Backscattering of dark photons into axions is essential to understand the final ALP relic abundance, however, capturing this non-linear effect requires simulating the system on a lattice. In this work, we present results of a lattice simulation of the axion-dark photon system on a $512^3$ lattice and obtain the resulting gravitational wave spectrum. Our formulation manifestly preserves the shift symmetry and gauge invariance of the continuum theory. We confirm the findings of Refs. [26–28] that the ALP relic abundance cannot be suppressed by more than about two orders of magnitude relative to the ordinary misalignment mechanism with no particle production.

For the GW signal, we find that the inclusion of backscatterings and GWs sourced from axion anisotropies broadens the spectrum towards the UV, while the peak frequency and amplitude are roughly consistent with the results from the linear analysis [14]. Furthermore, we find that the polarization of the GW spectrum now depends non-trivially on the coupling strength $\alpha$ and initial misalignment angle $\theta$. While the signal remains strongly polarized for smaller couplings, for $\theta\alpha \gtrsim 60$ the polarization is washed out due to backscatterings which couple the dark photon helicities. At even larger couplings, the polarization can flip from the initially dominant one and exhibit a non-trivial frequency dependence. If these features could be observed experimentally, they would provide additional information on the model parameters and potentially even the initial conditions after inflation.

As discussed in detail above, a large fraction of the parameter space of interest for experimental GW detection is inconsistent with the observed DM relic abundance. In Section 5, we sketch two simple extensions of the model that could potentially resolve this tension, which essentially come down to decreasing the axion mass after GW production, such that the experimental signatures remain unchanged. In the more radical approach, where the axion is rendered massless at late times, the dark photon can be given a small mass and play the role of dark matter. While much of the parameter space requires extending the model, a window remains for pulsar timing arrays to probe the original, minimal model.

### Acknowledgments

We thank Dani Figueroa, Naoya Kitajima, Jiro Soda, Yuko Urakawa and Zack Weiner for useful discussion. Additionally, the authors gratefully acknowledge the computing time granted on the Mogon supercomputer at Johannes Gutenberg University Mainz (hpc.uni-mainz.de). Work in Mainz is supported by the Deutsche Forschungsgemeinschaft (DFG), Project ID 438947057 and by the Cluster of Excellence "Precision Physics, Fundamental Interactions, and Structure of Matter" (PRISMA+ EXC 2118/1) funded by the German Research Foundation (DFG) within the German Excellence Strategy (Project ID 39083149).

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

## A    Scaling Relations

While the axion is pinned by Hubble friction, its energy is constant and dominated by the potential. Once the axion starts to oscillate around $t_{osc}$ defined by $H(t_{osc}) = m$, its energy and number density decrease as $a^{-3}$ until the backreaction from dark photon production becomes sizeable. In the regime of small misalignment angles $\theta \lesssim \pi/2$, where the quadratic approximation for the potential holds we find that the axion number density is well approximated as

$$n_\phi = \theta^2 m f^2 \left(\frac{a_{osc}}{a}\right)^3 , \tag{A.1}$$

after the onset of oscillation. Our analysis shows that the final abundance is suppressed by a factor typically of order $10^{-2}$ through the dark photon production as discussed above.

The abundance of dark photons is set during the initial backreaction at time $t_*$, when the majority of energy is transferred from the axion to the dark photon. Afterwards, it scales as

$$\rho_X = \rho_\phi\big|_{t=t_*} \left(\frac{a_*}{a}\right)^4 = \theta^2 m^2 f^2 \frac{a_*}{a_{osc}} \left(\frac{a_{osc}}{a}\right)^4. \tag{A.2}$$

The linear analysis describes the dynamics with great precision leading up to the backreaction and can be used to find an analytic estimate for $a_*/a_{osc}$. To do so, we assume that the energy in the dark photon is dominated by the fastest growing mode $k_* = \alpha|\phi'|/(2f) \approx \alpha\theta/2 \, (a_{osc}/a_*)^{3/2} a_* m$ and the energy in the dark photon is therefore given as $\rho_X^* \approx (k_*/a_*)^4 |v_*/v_{BD}|^2$, where $v_*$ is the dark photon mode function corresponding to $k_*$ at $t_*$ and $v_{BD}$ the Bunch-Davies mode function. Using the analytic estimate for the mode function $v_*$ found in Ref. [36], we can rewrite $\rho_X^* \approx \rho_\phi^*$ in the form of a transcendental equation for $a_*/a_{osc}$

$$\log\left(\frac{f}{\theta\alpha^2 m}\left(\frac{a_*}{a_{osc}}\right)^{3/2}\right) = \frac{\alpha\theta}{\sqrt{2}}\left(\frac{a_{osc}}{a_*}\right)^{1/4}$$

$$\times \left[0.6 - 0.82\sqrt{\frac{a_{osc}}{2a_*}} - 0.49\sqrt{\frac{a_*}{2a_{osc}}} + 0.45\frac{a_*}{a_{osc}} - 0.05\sqrt{\frac{a_*^3}{2a_{osc}^3}}\right]. \tag{A.3}$$

We compared these two equations to our results on the lattice and found that they track the scaling to within a factor 2 for $40 \leq \theta\alpha \leq 100$. For $\theta\alpha \gtrsim 100$ the backreaction occurs within the first period of oscillation and keeps the axion from efficiently rolling towards $\phi = 0$. This leads to a prolonged emission of dark photons that is not taken into account by these relations.

## B  Numerics

Below we summarize the details of our lattice implementation. For the axion and dark photon dynamics we closely followed Refs. [29, 30] while for the gravitational waves we adhere to Refs. [34, 35].

### B.1  Lattice Action

We use a staggered grid algorithm to solve the dynamics of the axion coupled to the dark photon. At the heart of these algorithms lies the notion of some fields lying between lattice sites. For example, the axion field, as it is parity odd, is displaced half a time step forward. We will denote this by $\phi(x + dx_0/2) = \phi|_{x+dx_0/2}$, where $x = (x_0, x_1, x_2, x_3)$ is a point on the lattice. Furthermore, we use a non-compact formulation of the $U(1)$ gauge dynamics, meaning that we use the field strength as our variable instead of Wilson lines. Since the gauge field $X_\mu$ is associated with the Wilson line linking neighboring lattice sites, it naturally is displaced by $+dx_\mu/2$ ($X_\mu|_{x+dx_\mu/2}$). We define the forward and backward derivative of a quantity $f(x)$ as

$$\Delta_\mu^\pm f(x \pm dx_\mu/2) = \frac{\pm f(x \pm dx_\mu) \mp f(x)}{dx_\mu}. \tag{B.1}$$

This reproduces the continuum derivative up to $\mathcal{O}(dx_\mu^2)$, but only if one expands around the natural lattice site $x \pm dx_\mu/2$ as one can easily check

$$\Delta_\mu^\pm f(x \pm dx_\mu/2) = \partial_\mu f(x \pm dx_\mu/2) + \mathcal{O}(dx_\mu^2). \tag{B.2}$$

The last rule needed for building the discretized version of the action in Eq. (2.1) is that the product of two operators that reproduce their continuum version up to second order is only of second order if the operators lie on the same lattice site.

We work in conformal time and assume that the contribution of the axion and dark photon to the total energy density is negligible, i.e. that the evolution of the scale factor is independent of the dynamics. We assume the scale factor is a given function $a(\tau)$ that can be evaluated to get $a|_\tau$ and $a|_{\tau+d\tau/2}$. The action we want to discretize reads:

$$S = \int d^4x \left[ \frac{a^2}{2} \partial_\mu \phi \, \partial_\nu \phi \, \eta^{\mu\nu} - a^4 V(\phi) - \frac{1}{4} X_{\mu\nu} X_{\alpha\beta} \eta^{\mu\alpha} \eta^{\nu\beta} + \frac{\alpha}{8f} \phi X_{\mu\nu} X_{\alpha\beta} \epsilon^{\mu\nu\alpha\beta} \right], \tag{B.3}$$

where $\partial_\mu = (\partial_\tau, \partial_{x_i})$ denotes the derivative with respect to comoving coordinates, $\eta^{\mu\nu} = \text{diag}(1, -1, -1, -1)$ is the inverse Minkowski metric, $X_{\mu\nu} = \partial_\mu X_\nu - \partial_\nu X_\mu$ is the dark photon field strength and $\epsilon^{\mu\nu\alpha\beta}$ is the totally antisymmetric tensor with sign convention $\epsilon^{0123} = 1$. The discretized version of the axion part of the action is

$$S \supset d\tau dx^3 \sum_x \left[ \frac{(a|_\tau)^2}{2} \Delta_0^- \phi \Delta_0^- \phi \Big|_x - \frac{(a|_{\tau+d\tau/2})^2}{2} \sum_i \Delta_i^+ \phi \Delta_i^+ \phi \Big|_{x+d\tau/2+dx_i/2} \right.$$
$$\left. + (a|_{\tau+d\tau/2})^4 \, V(\phi) \Big|_{x+d\tau/2} \right], \tag{B.4}$$

where we have indicated the exact lattice site of the displaced operators. The lattice version of the dark photon field strength is

$$X_{\mu\nu}|_{x+dx_\mu/2+dx_\nu/2} = \Delta_\mu^+ X_\nu - \Delta_\nu^+ X_\mu, \tag{B.5}$$

which is invariant under the gauge transformation

$$X_\mu \to X_\mu + \Delta_\mu^+ \alpha, \tag{B.6}$$

where $\alpha(x)$ is an arbitrary function of the lattice site. It is convenient to introduce the electric and magnetic fields as

$$E_i = X_{0i}|_{x+d\tau/2+dx_i/2} \tag{B.7}$$

$$B_i = \frac{1}{2}\epsilon_{ijk} X_{jk}|_{x+dx_j/2+dx_k/2}, \tag{B.8}$$

as this allows us to write the gauge kinetic term on the lattice as

$$S \supset d\tau dx^3 \sum_{x,i} \frac{1}{2}\left[ E_i E_i \Big|_{x+d\tau/2+dx_i/2} - B_i B_i \Big|_{x+dx_j/2+dx_k/2} \right]. \tag{B.9}$$

Finding a lattice version of the interaction piece is more challenging, since the electric and magnetic field strengths are associated with different sites on the lattice and therefore the first guess

$$S \supset d\tau dx^3 \sum_x \frac{\alpha}{f}\phi \sum_i E_i B_i, \tag{B.10}$$

does not reproduce the continuum action up to second order. The solution to this problem is to introduce averages of operators between lattice sites, since these reproduce the operator to second order on the site in between. In principle there are several averaging schemes, but one also needs to check that the shift symmetry $\phi \to \phi + \epsilon$ is respected and that the resulting equations of motion allow for an iterative solution. These issues are discussed in detail in Ref. [29], and we use the scheme found there, employing the following averages:

$$E_i^{(2)}|_{x+d\tau/2} = \frac{1}{2}\left( E_i|_{x+d\tau/2+dx_i/2} + E_i|_{x+d\tau/2-dx_i/2} \right) \tag{B.11}$$

$$\begin{aligned}B_i^{(4)}|_x = \frac{1}{4}\big( &B_i|_{x+dx_j/2+dx_k/2} + B_i|_{x+dx_j/2-dx_k/2} \\ &+ B_i|_{x-dx_j/2+dx_k/2} + B_i|_{x-dx_j/2-dx_k/2} \big).\end{aligned} \tag{B.12}$$

With these definitions, the interaction piece becomes

$$S \supset d\tau dx^3 \sum_x \frac{\alpha}{f}\phi \,\frac{1}{2}\sum_i E_i^{(2)}\big( B_i^{(4)} + B_i^{(4)}|_{+d\tau} \big)\Big|_{x+d\tau/2}. \tag{B.13}$$

## B.2 Equations of Motion and Integration Scheme

We work in temporal gauge where $X_0 = 0$. The dynamical degrees of freedom are $\Pi_\phi = \Delta_0^- \phi$ and $X_i$ given at time $\tau$ as well as $\phi$ and $E_i = \Delta_0^+ X_i$ at time $\tau + d\tau/2$. We use the defining equation of $E_i$ to find $X_i$ at $\tau + d\tau$

$$X_i\Big|_{x+d\tau} = X_i\Big|_x + d\tau \ E_i\Big|_{x+d\tau/2}. \tag{B.14}$$

By varying the action with respect to $\phi$ one finds the equation of motion

$$\Delta_0^+(a^2\Pi_\phi) = a^2 \sum_i \Delta_i^- \Delta_i^+ \phi - a^4 V'(\phi)$$
$$+ \frac{\alpha}{2f} \sum_i E_i^{(2)}\big(B_i^{(4)} + B_i^{(4)}|_{x+d\tau}\big) , \tag{B.15}$$

that is used to evolve $\Pi_\phi$

$$a^2(\tau + d\tau)\Pi_\phi|_{x+d\tau} = a^2(\tau)\Pi_\phi + d\tau \left[ a^2|_{\tau+d\tau/2} \sum_i \Delta_i^- \Delta_i^+ \phi \right.$$
$$\left. - a^4|_{\tau+d\tau/2} V'(\phi) + \frac{\alpha}{2f} \sum_i E_i^{(2)}\big(B_i^{(4)} + B_i^{(4)}|_{x+d\tau}\big) \right]. \tag{B.16}$$

Note that since $X_i$ is known at $\tau$ and $\tau + d\tau$, the calculation of $B_i$ and $B_i^{(4)}$ at these times is straightforward and the interaction term can be calculated explicitly. Now that $\Pi_\phi(\tau + d\tau)$ is known, $\phi$ can be evolved

$$\phi|_{x+d\tau\cdot 3/2} = \phi|_{x+d\tau/2} + d\tau\ \Pi_\phi|_{x+d\tau} . \tag{B.17}$$

Finally, we have the equation of motion of $X_i$ to evolve $E_i$

$$\Delta_0^- E_i = -\sum_{j,k} \epsilon_{ijk}\Delta_j^- B_k - \frac{\alpha}{2f}\left(\Pi_\phi B_i^{(4)} + \Pi_\phi|_{x+dx_i} B_i^{(4)}|_{x+dx_i}\right)$$
$$+ \frac{\alpha}{8f}(2 - d\tau\Delta_0^-) \sum_\pm \epsilon_{ijk}(2 + dx\Delta_i^+)\left(\Delta_j^\pm \phi\ E_k^{(2)}|_{x\pm dx_j}\right) . \tag{B.18}$$

Notice however, that the evolved $E_i$ appears not only on the left-hand side of the equation but also on the right-hand side in the interaction piece. Furthermore, the interaction piece features $E_i$ not only at different times but also at different spatial positions due to the averages, making an explicit solution impossible. We therefore use the following implicit method. First, we approximate the $E_i|_{x+d\tau\cdot 3/2}$ by the already known $E_i|_{x+d\tau/2}$ in the interaction piece to get

$$E_i|_{x+d\tau\cdot 3/2,1} = E_i|_{x+d\tau/2} + d\tau\left[ -\sum_{j,k}\epsilon_{ijk}\Delta_j^- B_k \right.$$
$$- \frac{\alpha}{2f}\left(\Pi_\phi B_i^{(4)} + \Pi_\phi|_{x+dx_i} B_i^{(4)}|_{x+dx_i}\right)$$
$$\left. + \frac{\alpha}{4f}\sum_\pm \epsilon_{ijk}(2 + dx\Delta_i^+)\left(\Delta_j^\pm \phi|_{x+d\tau/2}\ E_k^{(2)}|_{x\pm dx_j+d\tau/2}\right) \right]. \tag{B.19}$$

This first approximation only satisfies the equation of motion up to $\mathcal{O}(d\tau)$ and we therefore have to at least do one more iteration, where we use the $E_i|_{x+d\tau\cdot 3/2,1}$ we just found to

approximate $E_i|_{x+d\tau\cdot3/2}$.

$$
\begin{aligned}
E_i|_{x+d\tau\cdot3/2,2} = {} & E_i|_{x+d\tau\cdot3/2,1} \\
& + d\tau\Bigg[\frac{\alpha}{8f}\sum_{\pm}\epsilon_{ijk}(2+dx\Delta_i^+)\left(\Delta_j^\pm\phi|_{x+d\tau\cdot3/2}\,E_k^{(2)}|_{x\pm dx_j+d\tau\cdot3/2,1}\right) \\
& - \frac{\alpha}{8f}\sum_{\pm}\epsilon_{ijk}(2+dx\Delta_i^+)\left(\Delta_j^\pm\phi|_{x+d\tau/2}\,E_k^{(2)}|_{x\pm dx_j+d\tau/2}\right)\Bigg].
\end{aligned}
\tag{B.20}
$$

While this approximation is now correct up to $\mathcal{O}(d\tau^2)$, it still poses a violation to the shift symmetry $\phi \to \phi + \epsilon$. This violation can be suppressed via higher order approximations such as

$$
\begin{aligned}
E_i|_{x+d\tau\cdot3/2,n+1} = {} & E_i|_{x+d\tau\cdot3/2,n} \\
& + d\tau\Bigg[\frac{\alpha}{8f}\sum_{\pm}\epsilon_{ijk}(2+dx\Delta_i^+)\left(\Delta_j^\pm\phi|_{x+d\tau\cdot3/2}\,E_k^{(2)}|_{x\pm dx_j+d\tau\cdot3/2,n}\right) \\
& - \frac{\alpha}{8f}\sum_{\pm}\epsilon_{ijk}(2+dx\Delta_i^+)\left(\Delta_j^\pm\phi|_{x+d\tau\cdot3/2}\,E_k^{(2)}|_{x\pm dx_j+d\tau\cdot3/2,n-1}\right)\Bigg].
\end{aligned}
\tag{B.21}
$$

This concludes one time step in the evolution of the fields. To integrate the equations of motion, one repeats these steps. One can obtain one more equation of motion, the Gauss constraint, by varying the action with respect to $X_0$

$$
\sum_i \Delta_i^- E_i = -\frac{\alpha}{4f}\sum_i\sum_{\pm}\Delta_i^\pm\phi\,(2+d\tau\Delta_0^+)B_i^{(4)}|_{x\pm dx_i}.
\tag{B.22}
$$

Given the fields at the same times as in the begining of the step, checking this equation is straightforward, since $B_i|_{x+d\tau}$ can be calculated using Eq. (B.14). One has to choose the initial field configuration such that the Gauss constraint is fulfilled. Evolving the fields using the exact equations of motion then ensures that it stays fufilled at all times. It can therefore be used to check the accuracy of the implicit method solving for $E_i|_{x+d\tau\cdot3/2}$.

## B.3 Fourier Transformation and Polarization

We define the Fourier transformation of fields not spatially displaced from a lattice site (e.g. $\phi$ and $\Pi_\phi$) as

$$
\phi(\tau,\vec{k}) = \frac{L^{3/2}}{N^3}\sum_{\vec{x}}\phi(\tau,\vec{x})\exp\left(-i\vec{k}\cdot\vec{x}\right).
\tag{B.23}
$$

For fields that are spatially displaced, we take the displacement into account in the exponential. For example for the Fourier transform of $X_i$ which is displaced by $+dx_i/2$ we have

$$
X_i(\tau,\vec{k}) = \frac{L^{3/2}}{N^3}\sum_{\vec{x}}X_i(\tau,\vec{x}+dx_i/2)\exp\left(-i\vec{k}\cdot(\vec{x}+dx_i/2)\right).
\tag{B.24}
$$

Note that this also means that a field and its derivatives transform differently since the derivative is displaced. The benefit of this convention is that the relation between a field

and its derivative in Fourier space is simply

$$\mathcal{F}\big(\Delta_i^{\pm}\phi\big)(\tau, \vec{k}) = i \, p_i(k_i)\phi(\tau, \vec{k}) \,, \qquad p_i(k_i) \equiv \frac{2}{dx} \sin\left(\frac{dx}{2} k_i\right) . \tag{B.25}$$

Note that $p_i(k_i)$ is real, making the discussion of polarization easier as shown in Ref. [35]. It allows us to define the polarization with respect to the behavior under rotations around $\vec{p}(\vec{k})$, as in the continuum case

$$\sum_{j,k} \epsilon_{ijk} \, p_j(k_j) X_k^{\pm}(\vec{k}) = \mp i \, |\vec{p}(\vec{k})| X_i^{\pm}(\vec{k}). \tag{B.26}$$

## B.4 Initial Conditions

We investigate the process of particle production during a period of radiation domination, where the scale factor takes the form $a(\tau) = m\tau$. We start the simulation at $\tau_0 = 0.1/m$ when $H_0 = 100m$, such that the axion is pinned by Hubble friction and $\Pi_{\phi,0} = 0$, and assume the axion is displaced by $\phi_0 = \theta f$ from the minimum of the potential. The dark photon field is in the Bunch-Davies vacuum at the start of the simulation. This corresponds to $X_i(\tau_0, \vec{k})$ and $E_i(\tau_0, \vec{k})$ being drawn from a Gaussian distribution with widths $1/\sqrt{2|\vec{p}(\vec{k})|}$ and $\sqrt{|\vec{p}(\vec{k})|/2}$, respectively. Afterwards, the projector

$$P_{ij} = \delta_{ij} - \frac{p_i(k_i)p_j(k_j)}{|\vec{p}(\vec{k})|^2} \,, \tag{B.27}$$

is applied to ensure that the Gauss constraint Eq. (B.22) is initially fulfilled. We then take the inverse Fourier transform to arrive at $X_i(\tau_0, \vec{x})$ and $E_i(\tau_0, \vec{x})$.

## B.5 Lattice Dimensions and Number of Iterations

We choose the time step of our simulations as

$$d\tau = \frac{1}{4} \min\{dx, 1/(ma(\tau))\} \,, \tag{B.28}$$

in order to avoid instabilities as a result of the discretization. We varied the side lengths of the simulated volume $L$ and the number of lattice sites along each direction $N$ as well as the number of iterations used when implicitly solving for $E_i$. In Fig. 1, we show the evolution of the comoving number density of the axion and dark photon for a variety of choices for the above mentioned parameters. Except for the two runs where the length was chosen particularly small $L = \pi/(4 \cdot m)$, the results agree up to $\approx 10\%$ fluctuations.

Aside from the physical quantities we also monitored violations in the Gauss constraint (B.22). We introduce the quantity

$$\frac{\left\langle \left| \sum_i \Delta_i^- E_i + \frac{\alpha}{4f} \sum_i \sum_{\pm} \Delta_i^{\pm}\phi \, (2 + d\tau\Delta_0^+) B_i^{(4)}|_{x \pm dx_i} \right| \right\rangle}{\langle \sum_i |E_i| \, /dx \rangle} \,, \tag{B.29}$$

where $\langle ... \rangle$ denote averages over all lattice sites, to measure the relative error in the Gauss constraint. In Fig. 9 we show the evolution of this quantity. We note that the relative error

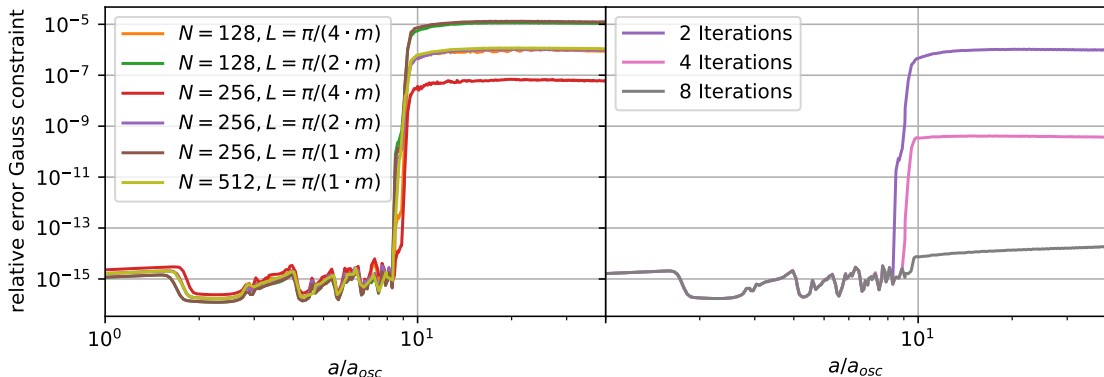

**Figure 9**. Evolution of the relative error in the Gauss constraint Eq. (B.29) for different choices of the number of lattice sites along each direction $N$ and the side length of the simulated box $L$ with fixed $\alpha = 60$, $\theta = 1$. The number of iterations used in the implicit scheme is fixed at 2 for the left panel while it is varied in the right panel with $N = 256$, $L = \pi/(2 \cdot m)$ held fixed. In all cases, the error stays close to machine precision $\approx 10^{-16}$ up to $a/a_{osc} \approx 8$, when the dark photon production backreacts on the axion. Thereafter, the error is minimized for small lattice spacings $dx = L/N$ and a high number of iterations.

starts out around $10^{-15}$ at $a = a_{osc}$ independently of the lattice parameters, close to the precision of a double precision float of $2^{-53} \approx 10^{-16}$. This goes to show that our procedure to initialize the dark photon indeed respects the Gauss constraint as expected. During the linear regime, when the dark photon energy is negligible compared to the axion, the relative error stays around $10^{-15}$ and only jumps up once the system enters the non linear regime, when the energy in the axion and dark photon becomes comparable and the axion field is fully inhomogeneous. As we can see from Fig. 9, the size of the violation of the Gauss error depends on the lattice spacing $dx = L/N$ and (as expected) on the number of iterations. It should be noted that already for $n = 8$ iterations the error in the Gauss constraint does not exceed $10^{-14}$ significantly and we expect it to stay at machine precision with only a few more iterations.

Since none of the physical quantities showed significant dependence on the number of iterations $n$ for $n \geq 2$, which is necessary to ensure convergence at $\mathcal{O}(dx^2)$, we set $n = 2$ for all the simulations discussed in the main text to minimize the computational effort. The choices for $N$ and $L$ listed in Section 3 were thus motivated by getting reliable results for the physical quantities, covering the relevant range of momenta and keeping computational costs down.

## B.6  Gravitational Waves

Following Ref. [34], we calculate the gravitational wave spectrum by solving for the transverse traceless (TT) fluctuations of the metric

$$\frac{1}{a^2}\partial_\tau(a^2\partial_\tau h_{ij}) - \nabla^2 h_{ij} = \frac{2}{M_P^2}\Pi_{ij}. \tag{B.30}$$

We note that this equation as well as the TT projection is linear, and for practical purposes we therefore solve

$$\frac{1}{a^2}\partial_\tau(a^2\partial_\tau\tilde{h}_{ij}) - \nabla^2\tilde{h}_{ij} = \frac{2}{M_P^2}S_{ij}\,,\qquad\text{(B.31)}$$

where $S_{ij}$ is the TT part of the energy-momentum tensor

$$S_{ij} = \partial_i\phi\partial_j\phi - \frac{1}{a^2}\left(E_iE_j + B_iB_j\right).\qquad\text{(B.32)}$$

The metric fluctuation $h_{ij}$ can then be obtained by applying the TT projection $\Pi$

$$\Pi(\tilde{h}_{ij}) = h_{ij}.\qquad\text{(B.33)}$$

From the source term we can immediately see that the corresponding fields on the lattice are not located on the same lattice site and an averaging scheme has to be employed. An important criterion when choosing this scheme, aside from practicality, is that it should allow for coherent interpretation of the TT conditions

$$\partial_ih_{ij} = 0,\qquad h_{ii} = 0.\qquad\text{(B.34)}$$

There exist many such schemes as discussed in Ref. [35]. Therein the authors find that the choice of scheme has only marginal influence on the results. In the scheme we employ, $h_{ij}$ sits at $x + d\tau/2$. Since the position of $h_{ij}$ is independent of $i$ and $j$, the trace can be calculated at each site $x + d\tau/2$. To find a local interpretation of the condition for transversality, we introduce the symmetric lattice derivative

$$\Delta_\mu^{\mathrm{sym}}\phi = \frac{\phi(x + dx_\mu) - \phi(x - dx_\mu)}{2dx_\mu}.\qquad\text{(B.35)}$$

The symmetric derivative reproduces the continuum derivative with $\mathcal{O}(dx_\mu^2)$ accuracy and is located at the same site as the field $\phi$ in contrast to the one-sided derivatives $\Delta^\pm$. With this, the transverse condition also takes a local form

$$\sum_i\Delta_i^{\mathrm{sym}}h_{ij}\Big|_{x+d\tau/2} = 0,\qquad \sum_ih_{ii}\Big|_{x+d\tau/2} = 0\,.\qquad\text{(B.36)}$$

The equation of motion on the lattice reads

$$\frac{1}{a^2}\Delta_\tau^-(a^2\Delta_\tau^+\tilde{h}_{ij}) - \Delta_k^-\Delta_k^+\tilde{h}_{ij} = \frac{2}{M_P^2}S_{ij}.\qquad\text{(B.37)}$$

Since the left side of the equation is located at the lattice site $x + d\tau/2$, we have to employ an averaging scheme such that $S_{ij}$ is located on the same site. To do so, we introduce

$$B_i^{(8)}|_{x+d\tau/2} = \frac{1}{2}\left(B_i^{(4)}|_x + B_i^{(4)}|_{x+d\tau}\right),\qquad\text{(B.38)}$$

and define on the lattice

$$S_{ij} = \Delta_i^{\mathrm{sym}}\phi\,\Delta_j^{\mathrm{sym}}\phi - \frac{1}{a^2}\left(E_i^{(2)}E_j^{(2)} + B_i^{(8)}B_j^{(8)}\right).\qquad\text{(B.39)}$$

With this explicit expression for the source term $S_{ij}$, Eq. (B.37) can be solved in a leap frog scheme to find $\tilde{h}_{ij}$. The associated momentum of the symmetric derivative is

$$\mathcal{F}\big(\Delta_i^{\mathrm{sym}}\phi\big)(\tau, \vec{k}) = i\, p_i^{\mathrm{sym}}(k_i)\phi(\tau, \vec{k}); \qquad p_i^{\mathrm{sym}}(k_i) \equiv \frac{1}{dx}\sin\left(dx\, k_i\right). \tag{B.40}$$

By replacing $\vec{k}$ in the continuum with $\vec{p}^{\,\mathrm{sym}}(\vec{k})$, the discussion of polarization and the TT projection is analogous to the continuum. Therefore, the two polarizations are defined by

$$\sum_{k,l} p_k^{\mathrm{sym}}(k_k)\big[\epsilon_{ikl}\, h_{lj}^{\pm}(\vec{k}) + \epsilon_{jkl}\, h_{il}^{\pm}(\vec{k})\big] = \mp 2i\, |\vec{p}^{\,\mathrm{sym}}(\vec{k})|\, h_{ij}^{\pm}(\vec{k}). \tag{B.41}$$