# Peer review of "Gravitational Waves from an Axion-Dark Photon System: A Lattice Study"

_SciPost Physics_

## Round 2 · Referee Report · Anonymous (Referee 1) · 2021-3-11

Strengths

1- New results that show a potential new source for gravitational waves that can be searched for in future experiments

2- Detailed study of the role of non-linear effects in this class of models

Weaknesses

1- Some of the results and methods were not discussed in a sufficiently clear way.

Report

The authors performed a careful numerical study of models in which axions produce a large abundance of massless dark photons through a tachyonic instability. They use the results of their simulations to compute the power spectrum of gravitation waves produced by the perturbations in the dark photon and axion fields, including the non-linear effects of the interactions between the fields and show the sensitivity of future detectors to the parameter space of the model. Their study produced new interesting results of broad interest to the community, in particular with respect to the gravitational wave spectrum, and improved the understanding of the effects of interactions in the tachyonic production of dark photons from axions. Their analysis seem sound and the journal's criteria for acceptance are clearly met. Nonetheless, I think there are a few points in which they should address in order to make their approach and results more clear:

  • All their lattice results used $m = 10^{-2}$ eV and $f = 10^{17}$ GeV. Their conclusions for other values of the parameters rely on the scaling relations discussed in Section 4.2. In that Section they find that there is only partial agreement between the naive dimensional analysis expectations, which they use to obtain the scaling relations, and results of the simulations. Can they provide either theoretical arguments or numerical evidence that support using the scaling relations besides what the authors present in the text?

  • In Figure 1 they show the normalized axion number density, is it defined somewhere in the text? Is that the total number density of axions or just of the homogeneous, zero momentum, component?

  • In pages 6 and 7, when they are discussing the features in Figure 2, they argue that when the ratio of redshifts is about 7 a new feature in both the dark photon and axion spectrum starts to arise, which is interpreted as being generated by the scattering of dark photons back into axions. Is there a simple way to estimate when these interactions become relevant without resorting to the simulation? This would make the connection between the new features and the scattering interpretation more transparent.

  • In Eq. 4.1, what is the definition of $\Omega_\phi^*$? Is there a way to easily present the ratio of the energy density in the scalar compared to the total energy density of the universe at the start of backreaction? (I am assuming that this is what $\Omega_\phi^*$ is related to?)

  • All of their results for the GW spectrum look very flat at low frequencies. Is this an artifact of their IR cutoff for the lattice? Because of that, it does not seem like the spectrum was close to approaching $k^3$ which is what they extrapolate it to at low frequencies. Is there an estimate of at what frequencies the behavior should approach this $k^3$ scaling?

  • At the end of Section 4 the authors compare their results to other recent numerical studies. They argue that they find significant differences, in particular with respect to the polarization of the GW signal. Do they have an understanding of where this difference might be coming from?

  • In Section 5 they provide some possible model extensions to decrease the final abundance of the axion and not be in conflict with cosmological measurements. One of the scenarios they investigate is one in which the curvature of the potential at small field values is smaller than at large field values, which effectively makes the mass of the axion in the early universe different than at late times when the field amplitude has decreased sufficiently. They show some results for the final abundance of that scenario in Figure 4, but they do not discuss how those results were obtained in detail. In particular, what was the lattice size that they used for these simulations? Given that now there is a smaller mass, would that require using a larger lattice in order to properly study the effects of the perturbations in the axion? Are there any new effects that arise from the parametric resonance associated with the perturbations of the cosine potential on top of the mass term? It would be interesting to see, even if in an appendix, more detailed results for that scenario.

Once the authors clarify these points, I believe this paper should be published in SciPost.

Requested changes

See report.

  • validity: high
  • significance: high
  • originality: high
  • clarity: good
  • formatting: excellent
  • grammar: excellent

Author:  Pedro Schwaller  on 2021-04-19  [id 1368]

(in reply to Report 1 on 2021-03-11)

We thank the referee for the positive evaluation and valuable feedback. We looked at the points that were raised and addressed them in the following way:

1) Since vacuum fluctuations of the dark photon (with energy of order m^4) are exponentially amplified to the energy scale of the axion, m^2 f^2, we expect m2f2 m4 exp(kt), where kis the most tachyonic scale. Thus, the number of e-folds between the start of oscillations and strong backreaction depends only logarithmically on f and m.

2) In Figure 1 we are plotting the volume-averaged number density, so it includes both the zero mode as well as fluctuations. We have clarified this point in the caption.

3) This process happens continuously but really only becomes important when the backreaction becomes strong around a/aosc = 11 and the axion field becomes highly inhomogeneous. Before this point, the axion field is approximately homogeneous and the dark photon power spectrum is dominated by a single polarization, which is the limit where the backscattering can be ignored.

4) All starred quantities indicate parameter values at the end of the perturbative phase, or when the backreaction becomes strong. Indeed, Omega_phi^ is the energy density fraction of the ALP at the time when the backreaction becomes strong. Because the background energy density scales as radiation, we have Omega_phi^ / Omega_phi^osc ~ a_* / a_osc, with Omega_phi^osc ~ m^2 theta^2 f^2 / (3H_osc M_P)^2.

5) Yes, this is an artifact of the fact that we can only make the box so large without losing resolution of the UV dynamics. We expect that the spectrum should approach k^3 scaling for modes that were outside the horizon at the time of backreaction, which would put the transition to k^3 scaling around k/m ~ (a_osc / a_*) ~ 0.1. Thus, our assumption of k^3 scaling after the IR cutoff of the lattice represents a conservative choice.

6) At this time we are unsure of the origin of the discrepancy, and the other recent numerical study does not provide sufficient lattice details for us to identify the origin of the discrepancy. However, a similar scaling of the GW polarization with alpha was found in 1805.04550 in the case of preheating after inflation.

7) As we note at the bottom of page 5 of the text, when attempting to capture the late time behavior of the axion abundance in Fig. 4, we used N = 128 and L = π/(2 · m) as the simulation must be run longer. The characteristic momentum of the process is set by the initial mass, so there is no need to change the lattice size in order to resolve the late time dynamics. As shown in 2004.07844, the condition to have a small mass at small field values also leads to a weak parametric resonance effect from the cosine term, so we do not expect any new effects there.

---

## Round 2 · Referee Report · Anonymous (Referee 2) · 2021-3-13

Report

The authors investigate the evolution of a pseudo-scalar field (an axion) coupled to a U(1) gauge field (a dark photon) through a Chern-Simons interaction term in the context of early-universe cosmology. The nonlinear dynamics of the system is studied with the aid of classical lattice simulations. The work also addresses phenomenological and experimental questions. The authors discuss in some detail what axion self-interactions could give rise to the right axion relic abundance for it to play the role of dark matter. The nonlinear axion-dark-photon system also acts as a source of primordial stochastic gravitational waves, and the authors study their properties numerically and discuss constraints on the model parameters by current and future gravitational wave experiments. I find the work to be a valuable contribution to the literature. It complements existing studies, which have not explored the dynamics of such systems in the model parameter range of this work. I recommend the manuscript for publication. I leave it to the authors discretion to address the following minor remarks.

1.) In section B.4, it is stated that $X_i$ and $E_i$ are drawn from a Gaussian distribution. Perhaps the authors may wish to clarify whether $X_i$ and $E_i$ are drawn independently or not. If these initial field fluctuations are meant to mimic the sub-horizon vacuum fluctuations of the dark photon field, then $X_i$ and $E_i$ are correlated and should not be drawn from the Gaussian distribution independently. One can see that by expanding the quantized $\hat{X}_i$ in terms of creation and annihilation operators. In going from the quantum $\hat{X}_i$ to the classical $X_i$ on the lattice, one simply replaces the creation and annihilation operators by complex numbers with an amplitude obeying the Rayleigh distribution and a uniformly distributed phase (see, e.g., arXiv:gr-qc/9504030 and documentation of arXiv:hep-ph/0011159). It is the same creation and annihilation operators which appear in the expression for the conjugate momenta $E_i$. That is why the same set of random numbers used to draw $X_i$ from its Gaussian distribution should be also used for $E_i$. This may not affect the results in the paper qualitatively, but in general affects the average over an ensemble of simulations with different initial fluctuations, and the authors should specify which approach to the initialization of the field fluctuations they used.

2.) The expression used to estimate the error in the Gauss constraint in Eq. B.29 contains volume averages in the numerator and denominator. The Gauss constraint should be respected at each lattice point, so perhaps a more appropriate estimator for the violation of the Gauss constraint is the maximum of the local ratio within the box, without taking volume averages, since in taking a volume average, there is the possibility of missing significant localized (i.e., occurring at a few points within the box) violations.

3.) The authors may wish to clarify that the exact equations of motion respect the discretized Gauss constraint in Eq. B.22 in the sense that the difference between the left and right hand sides of Eq. B.22 is conserved by the exact evolution. This difference is determined by the initial conditions and it cannot be set exactly to zero, due to the finiteness of the machine precision. This, together with the fact that not the solutions to the exact equations are used (since the system is implicit and can be solved only approximately), contribute to the violations of the Gauss constraint in Fig. 9.

4.) The authors have used the violation of the volume average Gauss constraint as an indicator for the number of iterations they need for their approximate implicit scheme for solving the equations of motion. Another important diagnostic that needs to be checked is the violation of energy conservation, and whether the used number of iterations allows for acceptable energy conservation. If the system was studied in a non-expanding spacetime (Minkowski), one would simply have to check the violation of the conservation of the volume average energy density $\dot{\bar{\rho}}=0$ and how it varies with the number of iterations. In a fixed power-law FRW background (like the radiation domination considered here), one can check instead the violation of the continuity equation $\dot{\bar{\rho}}+3H(\bar{\rho}+\bar{p})=0$ (see, e.g., documentation of arXiv:hep-ph/0011159). It will be helpful to the reader if the authors at least comment on the behavior of the violation of the energy conservation with varying the number of iterations of the implicit scheme.
  • validity: high
  • significance: ok
  • originality: ok
  • clarity: good
  • formatting: excellent
  • grammar: excellent

Author:  Pedro Schwaller  on 2021-04-19  [id 1369]

(in reply to Report 2 on 2021-03-13)

We thank the referee for the positive evaluation and valuable feedback. We looked at the points that were raised and addressed them in the following way:

1) Indeed $X_i$ and $E_i$ are drawn independently, and we have clarified this in Appendix B.4. Since the theory of a massless gauge field is conformally invariant, the modes experience no enhancement or 'squeezing' as referred to by arXiv:gr-qc/9504030 due to the expansion. The modes are only enhanced due to the tachyonic instability that arises once the axion starts freely oscillating. Since we initiate our simulation when the Hubble rate is 100 times the axion mass, we neglect this effect in the initial conditions, such that the modes are initially completely unsqueezed. As can be seen from Eq. 45 in arXiv:gr-qc/9504030, the amplitude of the modes $y(k)$ and their momenta $p(k)$ become uncorrelated in this limit, which corresponds to $F(k)=0$. Perhaps one might be worried that the classical description breaks down in this case, but since the transition is performed by replacing the operators by random variables, one can switch back and forth between the descriptions during the linear phase, since during this time all the physics is in the mode functions multiplying the operators/random variables. At the end of the linear regime, the mode functions are extremely squeezed and the transition to the classical description is valid.

2) The Gauss constraint can indeed be applied to each lattice point independently. We tested the referee’s suggestion of taking the maximum violation rather than the average and found that the new error estimate was one to two orders of magnitude larger than the average one. The new estimate fluctuated in the aforementioned range between each time step we evaluated it, leading to a broadening of the lines in Fig. 9. We added a footnote at the end of Appendix B.5 commenting on this.

3) As can be seen in Figure 9, the size of the jump in violation of the Gauss constraint heavily depends on the number of iterations, suggesting that this error arises from the usage of the implicit scheme. We tweaked the second paragraph of Appendix B.5 to clarify this point.

4) We checked the violation of the continuity equation as suggested by the referee. We found that it holds up to the level of a few percent. While there was a slight dependence on the lattice size and spacing it showed no dependence on the number of iterations. We added a comment in appendix B5 to this effect.

---

## Editorial Decision

resubmitted